# Theoretical and experimental analysis of circularly polarized luminescence spectro-photometers for artifact-free measurements using a single CCD camera

Bruno Baguenard[1], Amina Bensalah-Ledoux [1], Laure Guy [2], François Riobé [2], Olivier Maury[2] & Stéphan Guy [1]✉

Circularly polarized luminescence (CPL) is a fast growing research field as a complementary chiroptical spectroscopy alternative to the conventional circular dichroism or in the quest of devices producing circularly polarized light for different applications. Because chiroptical signals are generally lower than 0.1%, conventional chiral spectroscopies rely on polarization time modulation requiring step-by-step wavelength scanning and a long acquisition time. High throughput controls motivated the development of CPL spectrophotometers using cameras as detectors and space polarization splitting. However, CPL measurements imposes careful precautions to minimize the numerous artifacts arising from experimental imperfections. Some previous work used complex calibration procedure to this end. Here we present a rigorous Mueller analysis of an instrument based on polarizations space splitting. We show that by using one camera and combining spatial and temporal separation through two switchable circular polarization encoding arms we can record accurate CPL spectra without the need of any calibration. The measurements robustness and their fast acquisition times are exemplified on different chiral emitters.

The interactions between circularly polarized light and chiral material are unequal for left-handed and right-handed circular polarizations and are revealed by different phenomena such as optical rotation (refractive index difference), electronic circular dichroism (ECD) (UV-VIS absorption difference), vibrational circular dichroism (IR absorption difference) and so on[1]. The luminescence from chiral luminophore is also asymmetric in term of handedness which lead to the Circular Polarized Luminescence CPL = $\Delta I = I_L - I_R$ corresponding to the difference between the left-handed circular polarization (LHCP) and the right-handed circular polarization (RHCP) of luminescence[2]. CPL spectra of two enantiomers are opposite and the intensity of the signal is quantified by the dissymmetry factor $g_{lum} = 2(I_L - I_R)/(I_L + I_R)$. $g_{lum}$ is

of the order of $10^{-3}$ to $10^{-5}$ for most organic molecules[3,4]. CPL is a valuable complementary tool to the more standard ECD spectroscopy because it allows to access different electronic transitions and can be detected in non-transparent media[5,6]. During the last years, we observe a craze for CPL because of the synthesis of molecules or chiral systems with $g_{lum}$ higher than $10^{-3}$ making CPL a valuable alternative for the detection of chiral environmental probes or to conceive devices producing circularly polarized light[6–13]. Thus, the need for reliable, robust, and fast CPL measurements is becoming crucial. Most of the CPL spectrophotometers are based on a polarization modulation scheme: the polarization is modulated between the two LHCP and RHCP at some tens of KHz frequency with a photo-elastic modulator (PEM)[2].

[1]Institut Lumière Matière, Univ Lyon, Univ Claude Bernard Lyon 1, CNRS, F-69622 Villeurbanne, France. [2]Laboratoire de Chimie, ENS de Lyon, Univ Lyon, CNRS UMR 5182, F-69342 Lyon, France. ✉e-mail: stephan.guy@univ-lyon1.fr

The Fourier analysis of the signal allows to get the excess of circular polarization. This system requires a fast mono-channel detector (photo-multiplier or photodiode) with a time constant lower than the modulation period ($20\mu s$ for common PEM). Thus, it prevents the use of CCD camera which have, generally, millisecond-long integration times. The limit of detection is determined by the integration time of the lock-in amplifier used to demodulate the signal. The spectra are necessarily recorded step-by-step and the recording time may last several tens of minutes making this technique very heavy and limited to fundamental studies. This technique has proven its efficiency with measured spectra of $g_{lum} \leq 10^{-4}$ and its validity and related artifacts have been well investigated[14].

In this work, we investigate the use of a single CCD camera for a fast recording of full CPL spectra. Because, high-speed modulation can not be used (except for non-commercially available specialized camera[15]), CCD-based spectra must rely on a spatial polarization separation scheme. Spatial beams polarization splitting is commonly used for full polarization characterization, i.e., Mueller imaging. It is generally dedicated to single wavelength and/or high polarization dissymmetry ($\geq 10^{-2}$) phenomenon measurements, as complex procedures are required to calibrate each of the optical elements[16]. For measuring subtle spectral chiroptical signals, the use of a CCD camera after polarization beam splitting have been demonstrated for Raman optical activity[17] and CPL measurements[18–21]. The needed accuracy−of the order of 0.01% on the polarization difference for small organic molecules−requires equal control of the two analysing arms. This is almost impossible to achieve over the entire wavelength range and a calibration routine must be set up. The first strategy consists in measuring the calibration function between the two arms[18]. It is based on the division of one spectrum recorded on one of the polarization encoding arms by the same spectrum recorded on the other polarization encoding arm. This mathematical procedure leads to a dramatic noise enhancement of the calibrating function when the intensity of the denominator is close to the background noise and requires complex baseline removal in order to avoid "division by zero" errors. Nevertheless, it allows fast CPL measurements on high $g_{lum} \geq 0.1$ molecules but no validation on low $g_{lum}$ molecules has been described. On our side, we have also performed CPL measurements on a single camera by spatially separating the luminescence according to two polarization-encoded optical paths[21]. Recent works[22,23] also show that recording successively two polarization-encoded spectra is enough to extract the CPL of near-infrared emitters with $g_{lum} \geq 0.1$.

All these results have been obtained with emitters having CPL brightness factors $B_{CPL} \geq 0.5 M^{-1} \cdot cm^{-1}$[14]. In this study, we show that a single scan CPL setup can be extended for the measurement of CPL spectra of chiral compounds whatever their CPL brightness. At the cost of two successive measurements, corresponding to the swap of the two optical paths, and then their average, reliable spectra of CPL can be obtained without the need for calibration. This approach was already used to record Raman optical activity[17]. Here, the performances and limitations of this strategy for CPL measurements are theoretically and experimentally investigated.

The article is organized as follows. In the first part, after describing the experimental setup, we theoretically analyze the recorded spectra within the Mueller matrix framework. Three configurations are investigated: simultaneous measurements on two arms (spatial separation of polarization), successive measurements on one arm (time polarization's separation), and a combination of the two previous ones (space-time combination of polarization's separation). First-order calculations show that first-order false signals appear for spatial and time polarization separations. Most of those weak signals multiply together in the space-time combination and consequently, the related artifacts become very low. The only first-order remaining artifact comes from the polarizing beam splitter imperfections which transform luminescence linear anisotropy into false CPL.

These theoretical findings are experimentally illustrated in the second part of the paper, on broadband (~50 nm) with low $B_{CPL} = 10^{-4}$ $M^{-1} \cdot cm^{-1}$ and narrow line (~1 nm) with high $B_{CPL} \geq 10 M^{-1} \cdot cm^{-1}$ spectra of camphorquinone and $Eu^{3+}$ chiral complexes, respectively. Moreover, in order to assess the limits of our method and to verify its performance, we compare the spectra obtained with a camera to those obtained with our "homemade" single-channel PEM-based setup described elsewhere[7,24,25]. This work demonstrates that, for a solution without linear anisotropy, CPL free from first-order artifacts can be recorded in a few seconds over a wavelength range between 300 and 1050 nm with a multi-channel coverage of 150–300 nm depending on the grating and optical magnification. Depending on the CPL brightness, the integration time required for getting a signal-to-noise ratio higher than 10, ranges from 0.1 s to a few hundred seconds. The limitation coming from the mixing linear and circular anisotropies was also quantified in viscous fluorescein solutions. It was shown that a few percent of the linear anisotropy participates in the CPL signal due to the residual circular dichroism of the polarizing beam splitter. Moreover, the analog-to-digital converter of the CCD gives a fundamental limit of detection $g_{lum}$ ~$1.5 \times 10^{-5}$. The results described herein open perspectives on fast/automatic reliable measurement of CPL for compounds with $B_{CPL}$ as low as $10^{-4} M^{-1} \cdot cm^{-1}$.

## Results and discussion
### Principle and calculations

The homemade single CCD-based CPL spectrophotometer is schematically represented in Fig. 1. Basically, the handedness of the circularly polarized luminescence is spatially encoded into two geometrical paths before being spectraly dispersed by the spectrophotometer and recorded on the CCD camera. This is accomplished by the association of a quarter waveplate (QWP, fast axis 45°) and a polarizing beam splitter (PBS). Here we experimentally investigate CPL recording using time, spatial, and time–spatial polarization separation. For this, the QWP is held in a computer-controlled rotating holder. The measurement first involves setting the orientation of the QWP fast axis to +45° to place the RHCP channel on the top track of the CCD and the LHCP on the bottom track. These first two polarization-encoded spectra are recorded. In the second step, the fast axis of the QWP is set to -45° to invert the polarization channels and the two polarization-encoded channels are recorded again. This allows to measure the CPL in three ways:

- Spatial separation: $I_L$ and $I_R$ are simultaneously measured as the spectra coming from the two polarization-encoded paths for one given QWP orientation (Fig. 2a);
- Time separation: $I_L$ and $I_R$ successively measured on one optical path for two QWP azimuth $\pm \frac{\pi}{4}$ (Fig. 2b);
- Spatial-time combination: $I_L$ ($I_R$) is taken as the sum of the two left- (right-) handed polarization measurements for the two QWP orientations (Fig. 2c).

Ideally, all the configurations illustrated in Fig. 2 lead to the same true CPL spectra. However, since the optical elements along the two paths and the pump source are not ideal, we must consider different experimental limitations. The detailed steps to obtain the intensity of the recorded light are given in the Supporting Information. Here, we give only the major steps. Light is described using the four-components of the Stokes vector which are: the total intensity $S_0 = I_L + I_R$, the difference between the circularly polarized intensities $S_3 = I_R - I_L$ (opposite to the CPL), and the differences between the linearly polarized intensities $S_1 = I_0 - I_{90}$ and $S_2 = I_{45} - I_{-45}$. References axes (0° and 90°) are chosen identically to the PBS ones.

Using the Mueller matrices formalism, the Stokes vector at the detector can be derived as the product of the different Mueller matrices $\mathbf{M_i}$ representing each optical element.

We have calculated the four signals corresponding to the top and bottom arms measured twice with the QWP azimuth at $+\frac{\pi}{4}$ and

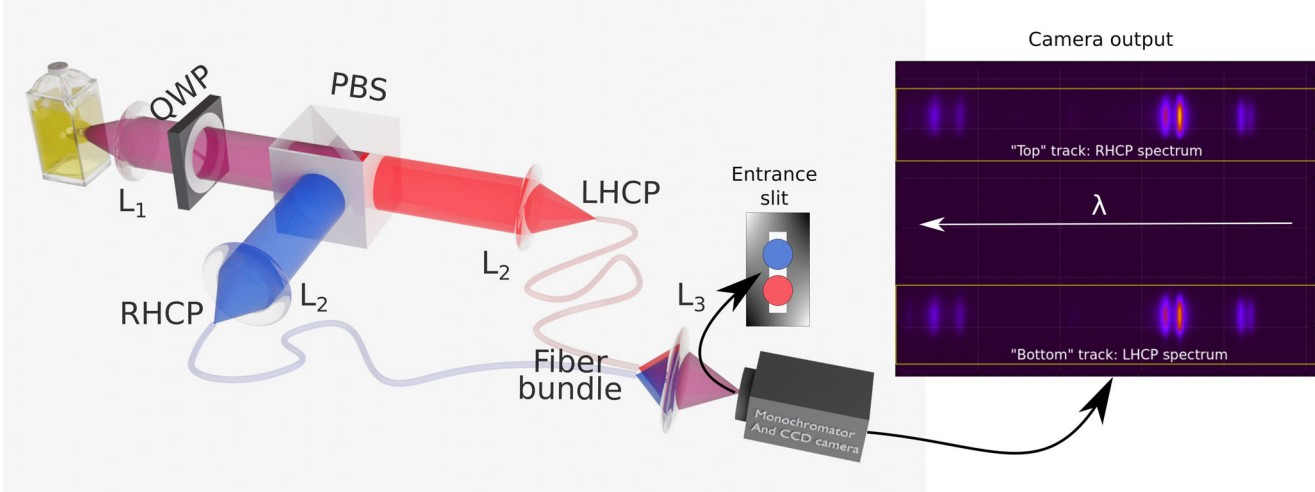

**Fig. 1 | Schematic of the CCD-based CPL spectrophotometer.** The fluorescence light is collected via lens $L_1$. After the quarter waveplate (QWP, main axis 45°) and the polarizing beam splitter (PBS, main axis 0°) the LHCP and RHCP are transmitted and reflected respectively (as horizontally and vertically polarized light). The two beams are launched into two fibers with a lens $L_2$, for each arm. These two fibers merge as a bundle. The bundle output is imaged via lens $L_3$ as two vertically separated spots on the entrance slit of the monochromator (see SI, Sec 1). The two spots are horizontally diffracted by the grating giving on the CCD camera the top/bottom spectrum coming from the top/bottom spots and thus related to the LHCP and RHCP emission. Here, the image is recorded from a $Eu^{3+}$ complex **2**.

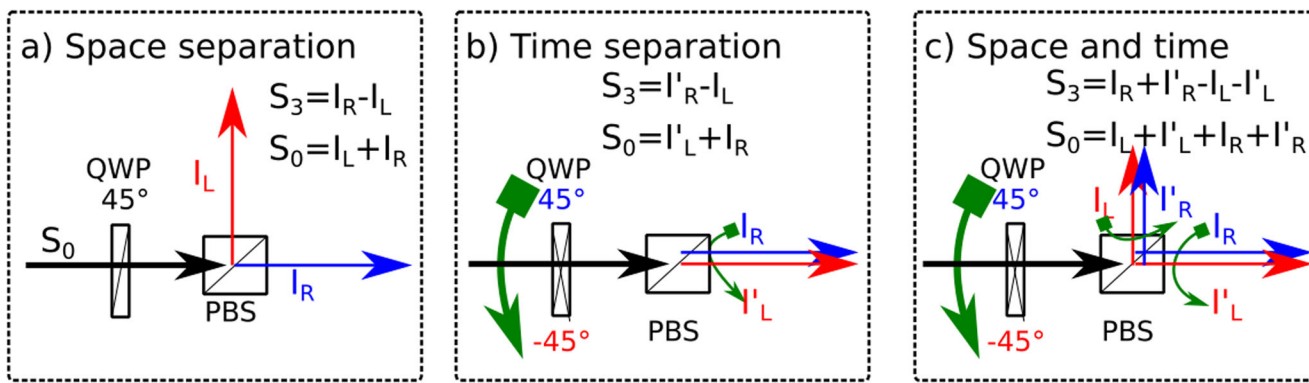

**Fig. 2 | Three measurement configurations to obtain CPL.** Left and right circular polarizations encoded spectra are measured: **a** simultaneously through the two arms, **b** on the same arm after rotation of the QWP, and **c** as the sum of the two previous ones.

at $-\frac{\pi}{4}$. Different experimental limitations have been taken into account: QWP phase retardation $\Psi \neq \pi/2$ and azimuth $\theta \neq \pm \pi/4$, the unequal transmission of the two arms, spectral mismatch, crosstalk between the two arms, unwanted PBS dichroism, and pump instability.

Several reasons make the overall transmission of the two arms unequal: imbalance of the beam splitter, unequal launching into the fibers, nonequal polarization response of the grating,...We have gathered all these imperfections in the transmission factor called $T_{\pm 1}$ for the top/bottom arms. $\Delta T$ and $\tilde{T}$ are the relative difference and average transmission of the paths.

The potential wavelength mismatch between the two spectra recorded simultaneously may come from a mis-alignment between the grating, the entrance slit, and the CCD pixel matrix. As a consequence, the two intensities recorded on the same vertical channel of the CCD correspond to the two slightly different wavelengths $\lambda_{\pm 1}$ for the top/bottom arms.

The cross-talk between the two polarization-encoded channels comes from the PBS imperfections and stray light inside the monochromator: a small relative part of one polarization falls into the detection area of the other one's. $\epsilon_{+,-}$ denotes the overall mixing from one channel to the other.

Our broadband PBS presents also unwanted linear ±45° and circular dichroism. Indeed, it transmits on each arm a few percents difference intensity for ±45° linearly or left-/ right-handed circularly polarized lights (Supplementary Fig. 3). These dichroism are taken into account in the Mueller matrix of the PBS as the $m_{02}$ and $m_{03}$ coefficients *denoted* $LD'_\pm$ *and* $CD_\pm$ where the underscript denotes either the transmitted or reflected beam. In the sum and subtraction measurements, these parameters appear as differences ($\Delta CD = CD_+ - CD_-$, $\Delta LD' = LD'_+ - LD'_-$) and average values ($\widetilde{CD}$ and $\widetilde{LD'}$).

Finally, because we measure spectra at two different times, a possible drift of the pump source may occur. We modeled this pump instability as two exciting ratio $\phi_{\pm 1}$ for the $\pm \frac{\pi}{4}$ QWP azimuth orientations. Again, we note $\Delta\phi$ and $\tilde{\phi}$ the relative difference and the averaged values of the pumping intensity with time, respectively.

All these sources with their quantification parameters are gathered in Table 1 and the corresponding Mueller matrices are described in SI sections 6 and 7. Taking into account all these effects, the spectra recorded for each arm following the two QWP

**Table 1 | The different defects of the setup limiting the experimental measurements, their origin, the parameters describing them, and their magnitudes**

| Defect | Origin | Parameter | magnitude |
|---|---|---|---|
| QWP retardance $\Psi \neq \frac{\pi}{2}$ | QWP specifications - chromaticity | $\psi = \Psi - \frac{\pi}{2}$ | $\pm \frac{\pi}{50}$ over 500 nm |
| QWP azimuth $\theta \neq \pm \frac{\pi}{4}$ | Misalignment | $\alpha = \theta - \frac{\pi}{4}$ | $\pm \frac{\pi}{200}$ |
| Unequal arms transmission: $T_{+1} \neq T_{-1}$ | PBS, fibers, grating | $\Delta T = \frac{T_{+1} - T_{-1}}{T_{+1} + T_{-1}}$ | $10^{-2}$ |
| Imperfect extinction ratio | PBS quality, stray light | $\epsilon_p$ | $10^{-2}$–$10^{-3}$ |
| PBS imperfection: LD', CD $\neq 0$ | PBS quality | LD'$_p$, CD$_p$ | $10^{-2}$–$10^{-3}$ |
| Pump source variations: $\phi_{+1} \neq \phi_{-1}$ | LED stability | $\Delta\phi = \frac{\phi_{+1} - \phi_{-1}}{\phi_{+1} + \phi_{-1}}$ | $10^{-2}$–$10^{-3}$ |
| Wavelength mismatch: $\lambda_{+1} \neq \lambda_{-1}$ | monochromator misalignment | $\Delta\lambda = \frac{\lambda_{+1} - \lambda_{-1}}{2}$ | 0.1 nm |

**Table 2 | CPL calculated at different orders of the experimental limitations using Equation (1) for three signal combinations**

| Polarization separation | Signals combination | ZO | First non-null correction | False CPL signals |
|---|---|---|---|---|
| Time: QWP (rotation) + PBS (One channel) | $I_{p,+1} - I_{p,-1}$ | $p(1 - \epsilon_p)S^e_{3(\lambda)}$ | $\Delta\phi S^e_0 CD_p S^e_1$ | Lum. Lin. Lum. |
| Spatial: QWP (fixed) + PBS (two channels) | $I_{+1,q} - I_{-1,q}$ | $q\eta_{mix}S^e_3(\lambda)$ | $\Delta T\, S_0 \Delta\lambda \frac{\partial S_0}{\partial\lambda}(-\psi + q\Delta CD)S^e_1 + (\Delta LD' + 2\alpha)S^e_2$ | Lum. Lum. deriv. Linear Lum. |
| Time + Space | $(I_{1,1} - I_{1,1}) - (I_{1,-1} - I_{-1,-1})$ | $-\eta_{mix}S^e_3(\lambda)$ | $\Delta CDS^e_1$ | Linear Lum. |

$S^e_{i=0..3}$ are the Stokes components of the emitted light. The p and q subscript denotes the polarization channel and the QWP orientation respectively. ZO is the result at zero-order. The first non-null order terms are in the fourth column. The last column shows the corresponding unwanted spectral signals present in the main signal.

azimuth orientations, written at first order:

$$I_{pq}(\lambda) = \frac{\tilde{T}\tilde{\phi}}{2}\left[(1 + p\Delta T + q\Delta\phi)S^e_0 + p\Delta\lambda\frac{\partial S^e_0}{\partial\lambda} + (-p\psi + qCD_p)S^e_1 \right.$$
$$\left. + (LD'_p + 2p\alpha)S^e_2 - (pq + q\Delta T + p\Delta\phi - pq\epsilon_p)S^e_3 - q\Delta\lambda\frac{\partial S^e_3}{\partial\lambda}\right] \quad (1)$$

where p = ±1 denotes the polarization channel after the PBS (+ and − for the horizontal and vertical polarization respectively) and q = ±1 denotes the orientations of the QWP fast axis (±1 for ±π/4). Therefore the pumping efficiency $\phi_{\pm1}$ is the pump efficiency at the time of the measurements.

For each of the three configurations, combining equation (1) with the respective (q, p) measurement pairs allows us to derive the experimental luminescence (by summation) and CPL spectra (by differentiation) (Supplementary sections 7.1-3). These calculations are performed as a powerful development with $S_0 \gg S_{i\neq0}$ and the experimental defects considered as small first-order effects (i.e., $\Delta T$, $\Delta\phi$, $\Delta\lambda\partial/\partial\lambda$, $\psi$, $CD_p$, $LD'_p$, $\alpha$, $\epsilon \ll 1$). The results are gathered in Table 2 for the CPL, the luminescence is given in Supplementary section 8. In Table 2, the measured signals are ordered following the zero and the first non-null terms of the power development.

As expected, the zero-order corresponds to the expected signals, i.e. CPL reduced by the factor $\eta_{mix}$ where:

$$\eta_{mix} = 1 - \frac{\epsilon_+ + \epsilon_-}{2} \quad (2)$$

quantifies the reduction of the CPL coming from the mixing between the two arms. The higher-order term comes from the experimental limitations that introduce unwanted signals mixed with the relevant ones. First-order terms in the luminescence signals are generally much smaller than the luminescence as they imply the product of small CPL and linear dichroism signals modulated by the instrumental defects. We can, therefore, safely assume that the sum signals are those counting for the luminescence.

In the first order, the measured CPL is scaled down by $(1 - \epsilon_p)$ or $\eta_{mix}$ for the time and spatial separation schemes respectively. CPL reduction coming from the PBS is about 1% and we measure in our

homemade monochromator a stray light coefficient lower than 4% over the whole spectral range. It gives an overall reduction factor of 0.95 on the measured CPL. The wavelength dependence of this factor is very smooth as it can be seen from the retardance and extinction ratio specifications given by different suppliers. Our own extinction ratio and stray light measurements validate the assumption (see Supplementary Fig. 2). Therefore, we do not take into account this correction in the rest of the paper.

The time separation of the polarizations is presented in the first row of Table 2. It induces two false CPL signals: $\Delta\phi \cdot S_0$ and $CD_p \cdot S^e_1$. The first one, $\Delta\phi \cdot S_0$, comes from the variation of the excitation source between the two $I_L$ and $I_R$ measurements. Indeed, a pump variation between the two measurements cannot be discriminated from a true variation of $I_L$ with respect to $I_R$. The second term, $CD_p \cdot S^e_1$, results from the transformation of linearly polarized light into circularly polarized one by the QWP. This circularly polarized light is more or less transmitted due to the residual circular dichroism of non-ideal polarizers. This term is only relevant if linear anisotropy is present in the luminescence as it could occur in anisotropic solid-state samples or via photoselection in isotropic solution[26].

These unwanted signals have the spectral shape of luminescence and can lead to an erroneous interpretation. They cannot be subtracted as a baseline because they are carried on by the signal itself. Therefore, designing a CPL apparatus requires that they are negligible compared to $S_3$. As $S_3 = g_{lum}S_0$, we obtain the requirements

$$\Delta\Phi \ll g_{lum}, \quad CD_p \cdot r_{lum} \ll g_{lum} \quad (3)$$

where $r_{lum}$ is the linear fluorescence anisotropy. Practically, it means that for standard $g_{lum} \approx 10^{-3}$ the measurement of CPL by temporally inverting the polarization requires stabilized excitation sources with a relative variation $\Delta\phi \leq 10^{-4}$. As the CPL is of low intensity, long integration time are needed and it is practically impossible to avoid the long-time excitation source variation except for the fast modulation techniques. Indeed, PEM modulation allows to invert the polarization selection at a very short time (a few μs) preventing any pump drift. Moreover, averaging over a large number of cycles increases the signal-to-noise ratio.

The constraint on the linear effect completely cancels for pure CPL emitters ($S^e_{1=0}$). However, if present, linear fluorescence anisotropy is seen as CPL scaled down by the factor $CD_p \ll 1$. Therefore, for systems containing both linear and circular anisotropies, the linearly polarized fluorescence must be characterized as well as the PBS imperfection (CD and LD') to ensure the true CPL measurement.

The spatial separation of the polarization is presented in the second row of Table 2. Simultaneous measurements over two polarization-encoded paths completely remove the pump variation effects but three other artifacts appear. The first one ($-S_0$) is related to the nonequal transmission between the two optical paths that can not be differentiated from a true nonequal $I_L$ over $I_R$ luminescence. Less intuitive, the wavelength mismatch of the two recorded spectra induces a CPL signal proportional to the luminescence derivative ($\frac{\partial S_0}{\partial \lambda}$). Indeed the difference between two *identical* spectra recorded with a wavelength mismatch $\Delta\lambda$ writes $\Delta S = S(\lambda + \Delta\lambda) - S(\lambda)$. Because $\Delta\lambda \ll \lambda$, we may use the Taylor-series expansion: $S(\lambda + \Delta\lambda) \approx S(\lambda) + \Delta\lambda \frac{dS}{d\lambda}(\lambda)$. Therefore, the recorded CPL signal, as a difference between two unmatched signals, contains unwanted terms proportional to the signal derivative. The last artifacts come from the mixing of apparatus defects (QWP and PBS) with the linear luminescence anisotropies. As three of them disappear after the time–spatial combination, we do not further discuss them at this point.

It is important to emphasize, that the artifacts described here are carried on by the fluorescence signal and its derivatives ($S_0$ and $\frac{\partial S_0}{\partial \lambda}$). Therefore, they add ambiguous contributions to the CPL with spectral features having similar shapes as the CPL itself. If the linear dichroism is neglected, the requirements for the experimental parameters in order to get signal artifacts lower than x% of $g_{lum}$ are :

$$\Delta T \leq x\% \cdot g_{lum} \quad \text{and} \quad \Delta\lambda \leq x\% \cdot g_{lum} \cdot \delta\lambda \qquad (4)$$

Here we use the fact that $\frac{\partial S_0}{\partial \lambda} \approx S_0 / \delta\lambda$ where $\delta\lambda$ is the luminescence spectral width. For example, at 10%$g_{lum}$ for small organic molecules ($g_{lum} \sim 10^{-3}$ and $\delta\lambda \sim 50$ nm), we get $\Delta T \leq 10^{-4}$ and $\Delta\lambda \leq 0.005$ nm. For a higher $g_{lum} \sim 10^{-2}$ observed in rare earth complexes ($\delta\lambda \sim 1$ nm), the constraints are $\Delta T \leq 10^{-3}$ and $\Delta\lambda \leq 10^{-3}$ nm. Whatever the sample, the required properties of the optical instruments controlling the transmissions better than 0.1% and the need for monochromators with sampling steps lower than 0.001 nm are unrealistic.

The first intuitive way to overcome the mis-balance between the two arms is to perform an intensity calibration between them. It is described in ref. [18] who divide the two signals coming from the two arms after inverting the role of one from left to right circular polarizer. Theoretically, it eliminates the sources of artifacts proportional to $\Delta T$. However, as it is based on the division of two signals, it induces high noise in the foot bands where the signal is close to the noise and special mathematical treatments are needed to avoid "division by zero" errors. Moreover, the derivative artifacts are enhanced as they appear both in the division and the subtraction of non perfectly $\lambda$ matched spectra and can explain the remaining artifacts discussed in this work. We show in the next section that the time-space combination is a much more robust method as it experimentally cancels the artifacts without mathematical signal division.

The combination of time and spatial configurations allows to reduce dramatically the artifacts related to the CPL measurement by our setup. Inspection of the first two rows of the Table 2 reveals that the first-order terms but CD do not depend on the $p$, $q$ parameters while $S^e_3$ (i.e., CPL) does. Consequently, all artifacts but CD add or subtract to the CPL depending on the two measuring schemes. Therefore, by adding the two CPL signals measured in the two configurations (last row of Table 2) the main remaining artifacts is the term $\Delta CD \cdot S^e_1$ and some second-order terms which are products of $\Delta\phi$ with

$\Delta T$ or $\Delta\lambda$. The requirements to avoid false CPL are then:

$$\Delta CD \cdot r_{lum} \leq x\% \cdot g_{lum}, \ \Delta\Phi \cdot \Delta T \leq x\% \cdot g_{lum}, \ \Delta\Phi \cdot \Delta\lambda \leq x\% \cdot g_{lum} \cdot \delta\lambda$$
$$(5)$$

The first term is relevant for linear fluorescence anisotropy and was already discussed for the time separation of the polarizations setup. For solutions of CPL emitters without linear anisotropy, only the two last terms of Eq. (5) remain. If spatial or temporal constraints alone are very difficult to maintain at very low level, the combination of spatial and temporal polarizations separations strongly reduces the effect of these constraints, rendering the experimental setup much easier to perform and thus reliable CPL measurements much easier to obtain.

A final important point is that the arm inversion must be done without changing the optical property of the arms except for the polarization encoding. It is therefore imperative not to change the polarization orientation just before the monochromator because the gratings are very polarization sensitive and with a complicated wavelength dependence. It is why we use a rotating waveplate followed by a fixed PBS: in this configuration, each detector has the same response whatever the type of polarization encoding and the autocompensation is valid. Inverted polarizing elements (fixed QWP and rotating polarizers[18]) results in an erroneous calibration due to the different polarization responses between the calibration and measurement stages.

## Experimental validation

Two kinds of molecular species have been used for this study: Camphorequinone **1**, a small organic molecule with a broad emission band and $g_{lum} \sim 10^{-2}$ (reference molecule used by the CPL spectrophotometer manufacturer: JASCO[27]) and rare earth chiral complexes with narrow emission lines in order to put into evidence the CPL artifacts related to fluorescence and its derivative. As these molecules present no detectable polarization photoselection, we do not discuss the linear fluorescence anisotropy in the following parts.

The luminescence and CPL spectra of **1** are displayed in Fig. 3. They present a broad single band extending from 400 to 550 nm. To compare their features, the spectra have also been recorded with our conventional step-by-step setup using a standard PEM plus analyzer system to differentiate via a lock-in amplifier the LHCP and RHCP. The spectra recorded with the two systems and corrected from the wavelength response, are very similar. The slight difference in the relative intensities of the fluorescence and CPL bands shows the residual error effects which are more visible in the case of wide emission bands, due to the different spectral responses of the two setups. As for each of them, the error on the fluorescence and CPL measurements is the same, it is canceled in the $g_{lum}$ spectra, as can be seen in Fig. 3. It took 2 h for the PEM-based setup against 10 min for the CCD-based one, to get spectra with equivalent signal-to-noise ratio (SNR).

The measured fluorescence and CPL spectra of $Eu^{3+}$ display the usual fingerprint with three main lines in the visible part of the spectrum (Fig. 4). Measurements using either the CCD (red line) or the step-by-step PEM (black circles) setups give the same results as expected. The acquisition time for the CCD setup is 12 s (1 s for the acquisition of each configuration plus 10 s time to rotate the QWP) against ~20 min for the mono-channel system, to measure equivalent spectra with $g_{lum}$ of a few tenths.

CPL in the near IR of enantiopure complexes **3** ([$Yb^{3+}$**(R,R)-L₃**] (OTf)₃ and [$Yb^{3+}$**(S,S)L₃**](OTf)₃ have been recorded using the spatial-time procedure, for the two enantiomers and compared with their counterparts obtained by our single-channel PEM-based setup with appropriate IR PM detector (Hamamatsu H10330B-75). The four luminescence spectra (Supplementary Fig. 5) are similar but to avoid crowding the graph, we only show one emission spectra for one

 

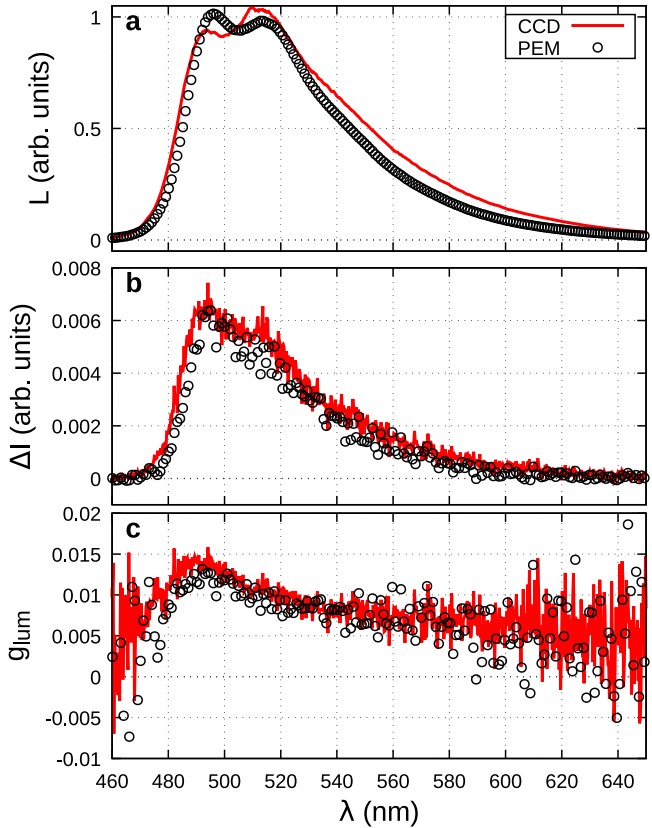

**Fig. 3 | CPL of camphorquinone.** Luminescence (**a**), CPL (**b**), and $g_{lum}$ (**c**) of (+)-**1** under 450 nm excitation. The scales luminescence and CPL are normalized to the maximum emission. In red the spectra were recorded with the CCD camera using the four signals procedure, in the black circle the spectra were recorded in a step-by-step setup using a PEM+analyser system. Wavelength response correction has been applied for each system.

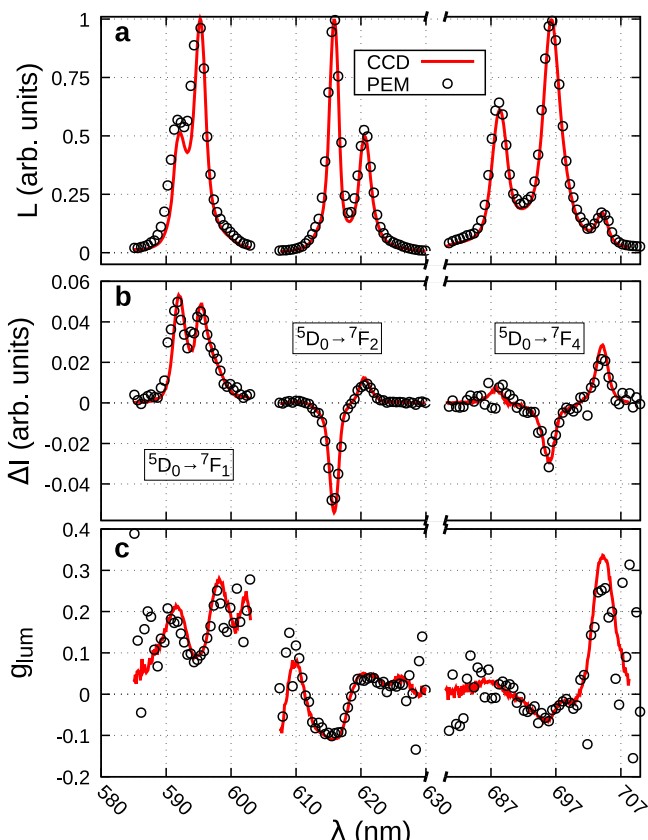

**Fig. 4 | CPL of Eu³⁺ complexes.** Three main visible transitions (luminescence (**a**), CPL (**b**), and $g_{lum}$ (**c**)) of Eu³⁺ complex **2** under 365 nm excitation. In red the spectra are recorded with the CCD camera using the time–spatial combination, in a black circle the spectra are recorded in a step-by-step setup using a PEM+Analyzer system. The scales of each luminescence and CPL transition are normalized to the maximum emission.

enantiomer, measured by each setup. The normalized emission spectra are almost identical and are typical of Yb³⁺. The difference in intensity at long wavelengths, despite their correction by the spectral response of the corresponding setup, is due to the detection limit of the CCD in this spectral range, inducing a high error scale. The corresponding CPL bands are well mirror images for the two enantiomers and for each of them, the spectra obtained by both setups are identical in terms of the number of bands, their shapes, and their positions in wavelength as well as their relative intensity. Even though the $g_{lum}$ value of the measured Yb³⁺-complexes is rather high (10⁻¹), the measurement of the CPL on the PEM side is more time demanding because of the low signal-to-noise ratio of the detector in this wavelength range. It took about 420 s (step = 0.5 nm with an integration time = 2.1 s per step) to record a CPL spectrum on the PEM side while only 10 s per spectrum (i.e., a global time of 30 s: 2 spectra + QWP rotation) was needed to obtain the same spectrum with, moreover, a better SNR, at the CCD side. Here again, our CCD-based setup with the spatial-time method shows its strength for the measurement of fast and reliable CPL in the near IR range. This range can be extended by the use of adapted CCD. It is worth noting that to date, only a few pioneering works concerning the measurement of CPL in the near IR region, using the conventional CPL spectrophotometer, have been reported[21,25,28,29]. IR region remains thus, a relatively new area to explore for CPL emitting materials and relative potential applications.

To check the linearity of the system, we prepared two enantiopure solutions of Eu³⁺ complex **2** at the same concentration of 10⁻⁵ M but with opposite handedness. We recorded a series of CPL spectra starting from one enantiopure solution and by adding,

gradually, the opposite enantiopure solution to the first one, we decrease the CPL of the mixture gradually until it is canceled for the racemic composition. The idea is to decrease the CPL signal gradually at a constant emission in order to characterize the linearity of the measurements. Supplementary Fig. 6 clearly shows that the decrease in intensity of the CPL follows the decrease in the enantiomeric excess of the solution. The inset in the figure attests to the linearity of our measurements.

SNR ≥10 are obtained for the Eu³⁺ complexes with an integration time of around 0.1 s. These molecules are high CPL brightness emitters with $B_{CPL} \in [10:300]\,M^{-1}\,cm^{-1}$ depending on the transition (Supplementary Table 3). This high CPL brightness associated with narrow lines allows sub-second acquisition. Lowering the $B_{CPL}$, increases the required integration time for getting the same SNR. For instance, Yb³⁺ complexes with $B_{CPL} \in [0.1:1]\,M^{-1}\,cm^{-1}$ require 10 s integration time to get SNR ≥10 (see Supplementary Fig. 5). The lowest CPL emitter we used is the camphorquinone **1** with $B_{CPL} = 6 \times 10^{-4}\,M^{-1}\,cm^{-1}$. The corresponding CPL spectra is displayed in Supplementary Fig. 7 for different integration times from 1 to 256 s using the CCD apparatus (this is the integration time for one QWP configuration). As expected, the SNR increases with the integration time, and this is without any artifacts coming from the lamp stability. To compare, we have used the same source and the same solution to record the CPL spectra with a step-by-step mono-channel setup (PEM plus analyzer). The overall experimental time of 480 s (i.e., a step = 0.25 nm and an integration = 0.7 s per step results in the black dotted spectra plotted at the top of the figure. The SNR is similar to what is obtained with the camera for 1s integration time spectra. For similar wavelength sampling intervals,

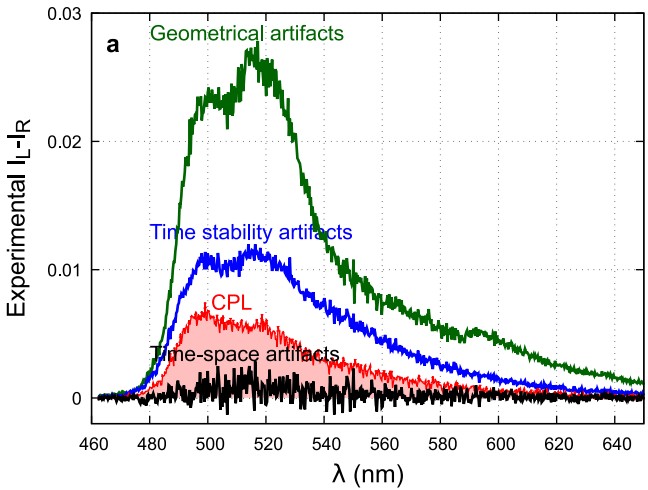
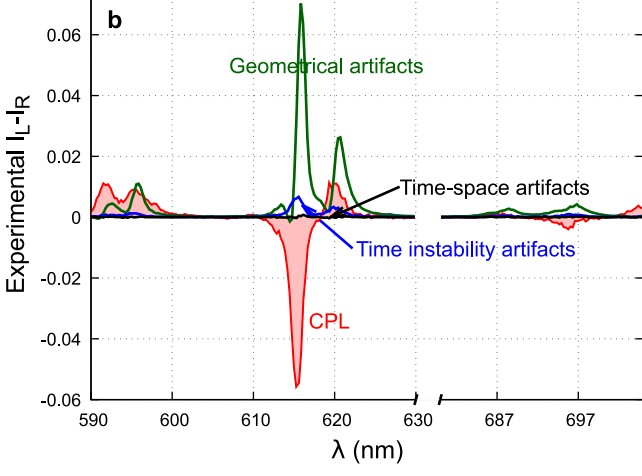

**Fig. 5 | Measured CPL artifacts.** CPL artifacts of **1** (**a**) and **2** (**b**) were recorded without the QWP (green-blue and black). To compare the CPL measured by the four signals combination is plotted in red. **1** is excited with a $\lambda = 450$ nm laser and 15 s integration time. **2** is excited with a $\lambda = 365$ nm emitting diode and 2 s integration time. Signals are normalized to the maximum of luminescence. The green curves are the signal difference between the two polarization-encoded arms: it images the unbalance of the optical paths. In blue, the signal difference for two measurements delayed by 20 s obtained on the same arm, this is the $\Delta\phi$ light source relative variation. 10 s represents the required time to rotate the QWP by 90°. Black curves result from the spatial-time combination.

the same spectral range and a comparable SNR, the recording with the camera is ~500 times faster.

### Robustness and artifacts

In order to characterize the measurement's artifacts with the CCD-based setup we proceed as follows. We first align our setup as well as possible. We take great care to make the two sensing paths as identical as possible (same lenses, same relative position of the optical elements, and top and bottom tracks centered on the camera). The spectrophotometer is also precisely aligned: slit, grating, and camera being parallel to each other, within one pixel. Then we measure the CPL spectra according to our spatial-time procedure (red line curves in Fig. 5). After the removal of the quarter wave-plate, we record exactly the same spectra. Assuming no linear fluorescence effects, we should obtain a zero CPL signal. However, because of the geometrical and spatial imperfections described in the theoretical part, the non-zero signal can arise. Mathematically speaking, taking off the QWP makes the $\mathbf{Q_q}$ (Supplementary Eq. 9) equals to the identity matrix. Therefore, because $S_1 = 0$ (no linear dichroism) only the $S_0$ term remains and the recording signals are the artifacts listed in Table 2 (fourth column).

Figure 5 displays the CPL artifacts measured on **1** and **2**. The difference between the two optical paths without QWP results in the green curve showing a false CPL similar to the luminescence spectra of **1** whose intensity is about 2.5% that of the luminescence spectra (Fig. 3). This is related to the optical unbalance of our setup. For the Eu$^{3+}$ complex, the recorded signal presents a false CPL opposite to the real one with an intensity of 7% of the luminescence (Fig. 4). This 2.5–7% of intensity variation is the optical limitation for the spatially separated polarization-based setups and only CPL with $g_{lum} \gg 0.05$ can be safely measured. The stability of the excitation source is given by the differential measurement on the same channel but at 20-s intervals (blue curve). Depending on the source, high power laser for **1** or low power light emitting diode for **2**, we observe intensity variations of about 1 and 0.5% respectively. These intensity variations come from the slowly varying drift of the pump source. To monitor this drift in order to scale the fluorescence intensity would require a detection setup with an accuracy $\approx 0.1\%$. This can not be performed with some standard photodiode power sensors which usually present a few percent accuracy. This is the limitation of time-separated polarization-based setups.

Finally, as described in the theoretical part, the combination of four measurements (black curve) through the time–spatial-based configuration dramatically decreases the artifacts down to $10^{-4}$ as the product of 2.5–7% times 0.5–1%. It allows the reliable measurements of samples with $g_{lum} \geq 10^{-3}$.

Next, we have experimentally investigated the robustness of the artifact-free CPL procedure. Starting from our best-aligned setup, we degrade on purpose some parts and record the corresponding spectra.

A variable neutral density filter in front of one of the two lenses $L_2$ (see optical setup Fig. 1) lowers the transmission along one polarization encoding path. The CPL spectra recorded in this configuration are displayed in Fig. 6 for **1** (left) and **2** (right). The measured CPL spectra using the four signals configuration are completely insensitive to the transmission difference between the two arms in the investigated range up to $\Delta T = 0.1$ for **1** and $\Delta T = 0.4$ for **2**. To compare, the CPL obtained for the spatially separated polarization are also displayed in the bottom panels. By looking at the luminescence spectra (Figs. 3, 4), it makes clear that the CPL artifacts depends on the luminescence as described by the term $\Delta T \times S_0$ (Table 2). In Fig. 6b, the false measurements are nearly proportional to $\Delta T$ with a shape very similar to the luminescence spectra of the corresponding molecule (Fig. 3) with some ripple effects attributed to the slow wavelength dependence of $\Delta T$. On the bottom right, the narrow emission lines only let appear the shape of the luminescence in the false CPL.

To add a small wavelength mismatch between the two polarization-encoded spectra, we tilted the camera a bit away from the well-aligned position. Again, we show that there is not any difference in the recorded CPL spectra using the time–spatial procedure (Fig. 7a, c). For this experimental misalignment, the CPL artifacts present in the spatial separated polarization procedure are related to the fluorescence derivative as illustrated in Fig. 7b, d. These artifacts are especially intense in the 620 nm region where the fluorescence derivative is the highest. At a maximum camera tilt angle of $-1.2°$, the artifacts are nearly proportional to the derivative (black dashed curve). By inverting the angle to $+1.2°$, the artifacts change in sign but not in shape as expected from the theory (Tab. 2, line 3: false CPL $\propto \Delta\lambda \frac{\partial S_0}{\partial \lambda}$). The order of magnitude of the measured CPL compare to the luminescence derivative for the Eu complex (Fig. 7d), shows that for our best setup (red curve) the wavelength mismatch is negligible around 595 nm and is about 0.02 and 0.1 nm at 620 and 700 nm respectively. It indicates that, under the best alignment, we can not guarantee a perfect spectral

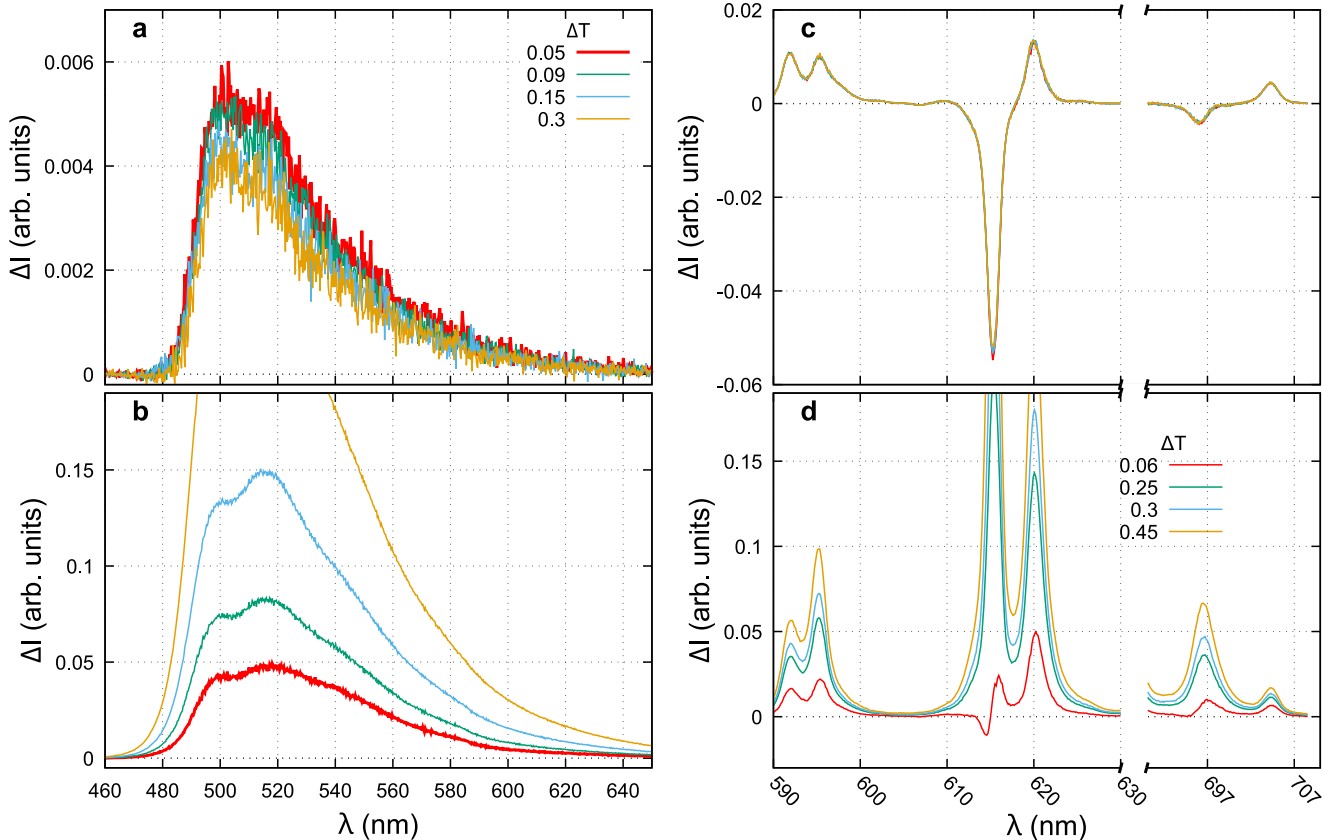

**Fig. 6 | Artifacts coming from arms imbalance.** CPL of **1** (**a**, **b**) and **2** (**c**, **d**) by using either the artifacts-free procedure (**a**, **c**) or taking one single $I_L$-$I_R$ difference (**b**, **d**). Note the y-axis different range between the top and bottom. The different curves are recorded for different neutral density attenuation on one arm. The average transmission difference factor $<\Delta T>$ is calculated from the luminescence integral measured at each arm.

matching on the whole spectral range (here 120 nm). Moreover, as the spectral sampling interval is 0.13 nm, it is not possible to get better accuracy and the CPL artifact related to this wavelength mismatch between the two recorded spectra on two different parts of the CCD is inevitable. However, our procedure is very stable in terms of CPL results with reproducible spectra even with strong misalignment (see the image distortion obtained for a 1.6° tilt in Supplementary section 9).

Finally, we investigated the effect of linear anisotropy by photoselection on our setup. For this purpose, we recorded the CPL and the linear polarization excess of a solution of fluorescein in a highly viscous water solution. Fluorescein is a well-known achiral molecule showing linearly polarized fluorescence due to photoselection[26]. As achiral molecules, it must be CPL silent. Emission was collected at 90° to the excitation beam which is either vertically or horizontally polarized. The water viscosity and therefore the photoselection was increased by adding sucrose in the water up to 1.7 w/w. The linear anisotropy (difference between vertically and horizontally polarized emission corresponding to the Stokes $S_1^e$ parameter) was measured by replacing the QWP with a half-waveplate. The $S_1^e$ and CPL signals are presented in Supplementary Fig. 8. As expected, when the sample is excited with a horizontally polarized light neither linear nor circular differential signals are detected. When the solution is excited by a vertically polarized light, linear anisotropy is detected. $S_1^e = 0.02$ for pure water solution and nearly reach $S_1^e \sim 0.3$ for the highly viscous solution (top panel). For $S_1^e = 0.02$, no CPL is detected by the apparatus. However, for higher linear anisotropy, a clear false CPL signal is detected. The ratio of linear and circular anisotropy spectra reveals that the CPL signal is about 4% of $S_1^e$. Our CD measurements of the PBS show CD spectra in the order of a few percent for the reflected and

transmitted beams (Supplementary section 4). This is in good agreement with our theoretical analysis which gives a linear-CPL mixing proportionally to ΔCD.

The CPL measurement by combining four measures corresponding to the spatial and temporal separation of the circular polarizations allows a self-compensation of the artifacts and the direct measurement of $S_3$ to the nearest $\eta_{mix}$ (Table 2, third column) with $\eta_{mix}$ defined in Eq. (2). ΔT vanishes at first order via the time–spatial measurement combination. It is not a critical point as experimentally demonstrated in Fig. 6. The QWP phase retardation $\Psi \neq \frac{\pi}{2}$ and azimuth $\theta \neq \pm \frac{\pi}{4}$ also cancel at first order. They are not critical. QWP azimuth accuracy around 0.1° can be easily achieved using standard alignment procedure by placing the QWP between two cross polarizers. Even if this orientation is not accurate, a misalignment of ±2° results in an error of 0.06%. Therefore, the standard quality of the QWP and its rough orientation does not induce significant errors in the CPL magnitude. Contrary to the QWP, the quality of the PBS or any polarizer used in a CPL measurement setup is very important. If the polarizer extinction ratio is 90%, for example, then 10% of one polarization is injected in the arm transporting the other one. This results in an underestimation of the CPL of 20%. Thus, the better the polarizer, the lower the error on CPL. Besides, a small relative part of one polarization falls into the detection area of the other ones depending on the stray light of the spectrophotometer. This leads again to an underestimation of the CPL of 2 to 4% in our setup. The stray light is the limit of CPL measurement setups based on a single spectrophotometer. The use of two spectrophotometers improves this aspect but will introduce a new source of errors due to the responsivity drift between the two detectors. The quality of the polarizer is also a key factor to avoid linear fluorescence anisotropy contribution in the CPL measured signal. The standard cube beam

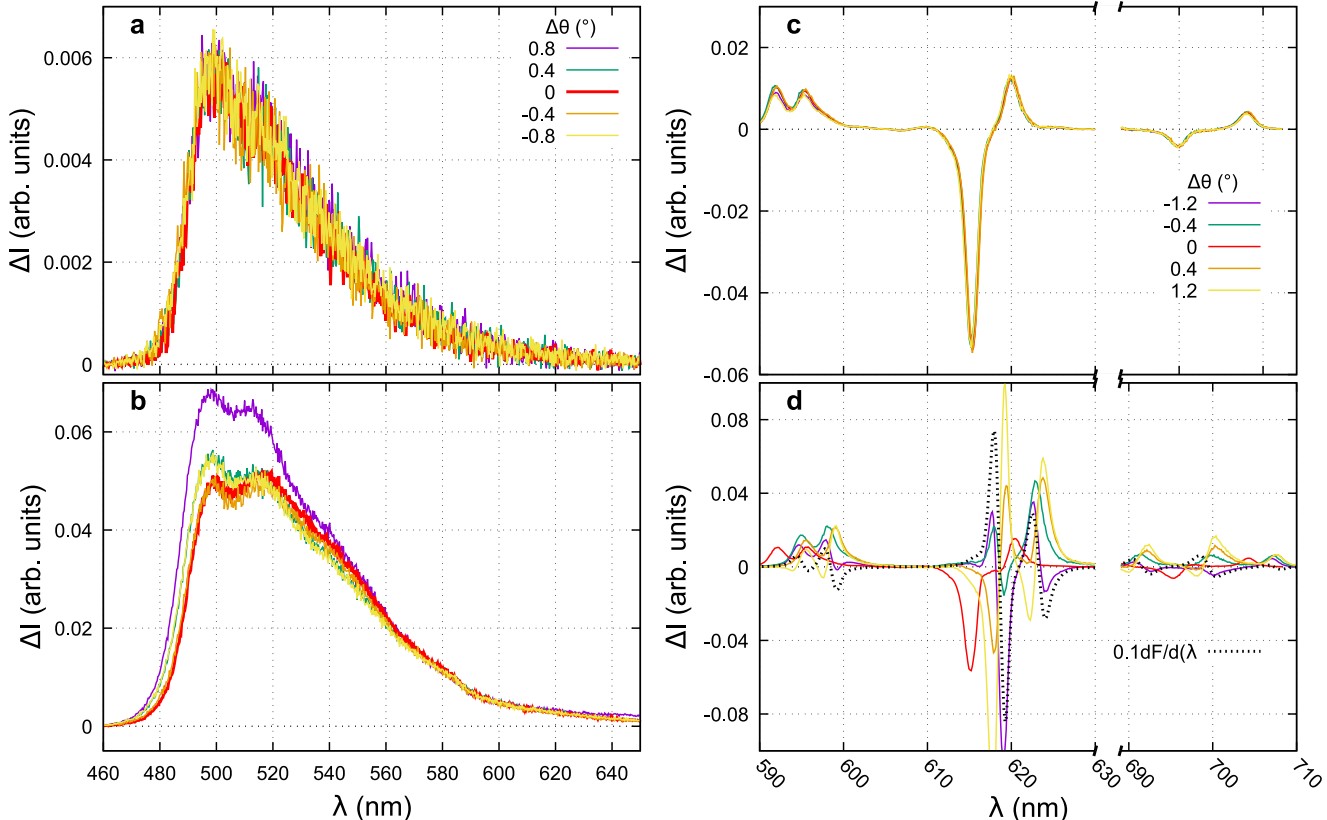

**Fig. 7 | Artifacts coming from wavelength mismatch.** CPL of **1** (**a**, **b**) and **2** (**c**, **d**) by using either the artifacts-free procedure (**a**, **c**) or taking one single $I_L$-$I_R$ difference (**b**, **d**). The different curves are recorded for different tilt mis-alignments of the CCD camera. Note the y-axis scale difference between the top and bottom panels. The dashed black curve in panel **d** is the derivative of the luminescence signal.

splitter used for this work presents CD signals of a few percent limiting this setup in the study of isotropic solution without anisotropy photoselection. Better quality polarizers such as Wollaston prisms should be used for samples mixing circular and linear anisotropies. However, they would require more complicated optical schemes as the deviation angle of the two separated beams is wavelength dependent.

The artifact sources for the three configurations of CPL measurements have been identified. The theoretical analysis of these artifacts shows that their spectral shape is proportional to the fluorescence signal and to its derivatives and therefore, cannot be subtracted as a simple baseline. Besides, their experimental quantification, by removing the QWP and measuring each of them, allows to assess the limits of each configuration.

Mixing of linear and circular anisotropies comes from imperfections of the polarizing elements (QWP retardance and orientation as well as PBS residual CD and LD'). These mixing are maximum for the spatial separation while they reduce to the PBS CD and the differential CD for the time and time-space configurations respectively.

Artifacts related to the spatial separation of the polarizations i.e., nonequal transmission ($\Delta TS_0$) and wavelength mismatch ($\Delta\lambda \frac{\partial S_0}{\partial \lambda}$) between the two optical paths (Table 2, second row) can not be suppressed in the case of CPL measurements using only spatial separation. The constraints on $\Delta T$ and $\Delta\lambda$ are CPL dependent ($\Delta T \ll g_{lum}$; $\Delta\lambda \ll g_{lum} \cdot \delta\lambda$) and require unrealistic control of the transmissions, better than 0.1% and spectrophotometer sampling step, lower than 0.001 nm. Experimentally, with the best alignment of our setup, the unbalance between the two arms induced not only false CPL whose intensity is about 2.5−7% that of the luminescence, but also a CPL band opposite to the real one, in the case of narrow emission bands (Fig. 5). Indeed, the artifacts related to the wavelength detuning are proportional to the derivative of the luminescence signal and therefore more important for the narrow emission lines. They introduced a false CPL signal with an intensity of ~8% that of the luminescence. Hence, in order to avoid as much as possible these false CPL contributions adapted and rigorous calibrations are needed and have to be checked before each set of measurements.

These constraints are no longer relevant in the case of CPL measurements by temporally inverting the polarization, as only one optical path is involved. However, to get non-erroneous CPL spectra, the excitation source stability is crucial with its variation between the two $I_L$ and $I_R$ measurements, $\Delta\Phi \ll g_{lum}$. This stability is difficult to ensure during the two successive measurements, especially since long integration times are necessary for accurate CPL measurements. Using a LED excitation source with the long-term stability of a few percent and standard optical elements, a false CPL signal whose intensity is equal to 1% of the luminescence, was measured.

By combining spatial and temporal separation of the circular polarization and performing two sets of measurements where the role of the polarization-encoded arms are inverted, then combining the four obtained measures, the first-order artifacts relative to the two precedent configurations vanish and only second-order terms remain (see Table 2, third row). Apart from the CCD calibration for wavelength accuracy, fast CPL spectra can then be safely recorded by the CCD-based spectrophotometer without the need for tedious alignment and heavy calibration procedure each time a sample or an optical element is changed.

Compared to the standard modulation technique with a lock-in amplifier, CCD-based devices do not have an AC filter. Therefore, the intensity resolution is the same over the whole range of the measurement and small signals are difficult to extract. In our case, with a 16 bits camera, the analog-to digital conversion leads to an accuracy of

$\frac{1}{2^{16}} = 1.5\,10^{-5}$ for a full signal range. This is the fundamental lower limit for $g_{lum}$.

To resume, we studied the limitations of three different configurations for a CPL spectrophotometer: time, spatial and spatial-time separations of the polarization. We demonstrated that unless using a rigorous and heavy calibration procedure, the artifacts related to the two first configurations cannot be completely suppressed and thus, inevitably lead to more or less false CPL spectra. This can range from a wavelength shift, deformation, and false relative intensities to a significant reversal of the CPL bands. Consequently, a misinterpretation of stereochemical structures and associated transitions of the studied chiral system or a wrong evaluation of the $g_{lum}$ is very likely. However, we showed that the third configuration, i.e., the spatial-time combination is the most efficient for obtaining reliable CPL spectra capable of measuring $g_{lum} > 10^{-4}$ in a robust, reproducible, and fast way. It offers many advantages over CPL measurement by only time or only spatial separation of the polarization. First, accurate and fast measurements with standard optical elements for $g_{lum} \geq 10^{-3}$ are obtained. The integration time targeted for an SNR higher than ten ranges from 0.1 s for $B_{CPL} \geq 10\,M^{-1}\,cm^{-1}$, narrow band emitters to a few hundreds for low $B_{CPL} \sim 10^{-3}\,M^{-1}\,cm^{-1}$ broadband ones. Moreover, the procedure does not rely on calibration, the measured signal is directly the CPL times a correcting factor between 0.92 and 1 coming mostly from the residual stray light and the PBS imperfections. The artifacts auto-compensate at first order without the need for high-quality QWP. Finally, the remaining artifacts can be measured on the same sample by just taking off the QWP.

Compared to a more standard mono-channel setup with polarization modulation, the recorded time is reduced by three orders of magnitude, for the same signal-to-noise ratio, and a fast measurement on a whole spectral range is obtained in one shot. The auto-calibration procedure allows recording of satisfactory spectra not only for high $B_{CPL}$ emitters as previously published but also for very low $B_{CPL}$ emitters as the camphorquinone. The cost of this kind of apparatus is about 40k€ mainly driven by the CCD camera (about 30k€). The implementation of such a fast and robust CPL spectrophotometer opens interesting perspectives for the monitoring of dynamic processes such as chemical reactions that vary over time, depending on a particular parameter. Moreover, thanks to its rapid measurements, the stabilization of the external parameter, such as temperature or magnetic field, is not anymore as critical as in the case of much longer measurements with PEM-based setups.

## Methods

### Optical setup

The homemade single CCD-based CPL spectrophotometer is schematically represented in Fig. 1. Basically, the handedness of the circularly polarized luminescence is spatially encoded into two geometrical paths before being spectrally dispersed by the spectrophotometer and recorded on the CCD camera. This is accomplished by the association of a quarter waveplate (QWP, fast axis 45°) and a polarizing beam splitter (PBS): the left-handed circularly polarized (LHCP) light is transmitted by the PBS (0° linear polarization) while the right-handed circularly polarized (RHCP) light is reflected by the PBS (90° linear polarization). The two-handedness-encoded beams are then routed at the adjustable entrance slit of the spectrophotometer by means of a dual-core fiber bundle (200 μm diameter). The two beams are spatially imaged as two vertically aligned spots on the entrance slit. Inside the spectrophotometer (see SI, Sec 1 for details), the input slit is imaged on the CCD camera (Andor iDus-420) after being horizontally diffracted by the grating leading to two vertically split spectra: the top (bottom) one corresponds to the top (bottom) fiber and consequently to the RHCP (LHCP) luminescence.

Our setup can operate in a wide wavelength range from UV to near IR. Indeed, by changing only the PBS cube and the transmission grating, we can easily switch from the UV-visible to the near-infrared spectral range. All other optical and detection elements (QWP, lenses, fibers, and CCD) are chosen to cover the entire spectral range from 300 nm to 1.1 μm.

### CPL measurements

**1** have been purchased from TCI company in enantiopure form, 1(R)-(-)-Camphorequinone and 1(S)-(+)-Camphorequinone (Supplementary Fig. 9). Fluorescence and CPL measurements have been carried out on solution samples by dissolving these molecules in ethanol. Typical concentrations of $8 \times 10^{-3}\,mol\,L^{-1}$ were prepared. For the CPL measurements of **1**, with $g_{lum} \sim 10^{-2}$, we used an 830 gr/mm grating and a 100 mm focal lens leading to a dispersion over a 270 nm spectral range and a wavelength sampling interval of ~0.3 nm, at the CCD side. The resolution was 1 nm and the integration time to get the spectra was 10 min (5 min for the acquisition of each configuration plus 10 s time to rotate the QWP). Camphorequinone solutions were excited with a laser diode (10 mW, $\lambda = 450\,nm$). The spectra have also been recorded with our step-by-step setup using a standard PEM plus analyzer system to differentiate via a lock-in amplifier the LHCP and RHCP. In this case, the spectral resolution was 1 nm, the step was 0.25 nm and the integration time was 7 s/step.

Complexes **2** and **3** are isostructural complexes of $Eu^{3+}$ (**2**) and $Yb^{3+}$ (**3**), respectively. They were prepared in their enantiopure forms ($[Ln(\mathbf{R,R})\text{-}\mathbf{L}_3](OTf)_3$ and $[Ln(\mathbf{S,S})\text{-}\mathbf{L}_3](OTf)_3$), according to the procedure we previously described[21]. Their structures (see Supplementary Fig. 9) was shown identical in both solid state and solution and their high brightness makes them good candidates as a reference for the calibration of CPL setups.

For the measurements of complexes **2**, an 830 gr/mm grating and a 200 mm focal lens was used providing a spectral range and a wavelength sampling of ~135 and ~0.15 nm, respectively. Spectra were acquired under UV LED (4 mW, $\lambda = 365\,nm$), with an equivalent resolution of 0.5 nm and acquisition time of 1 s for each spectrum at the CCD side and 2.1 s/step at the PEM one.

The spectra of **3** are recorded over a 940–1040 nm spectral range, under 405 nm laser excitation (power ~10 mW). On the PEM side, the spectra were recorded with a step of 0.5 nm and an integration time of 2.1 s/step. At the CCD side, a 300 gr/mm grating and a focal lens of 200 mm was used providing a spectral range and a wavelength sampling of ~350 nm (larger than the whole $Yb^{3+}$ fluorescence range) and ~0.35 nm, respectively. The resolution was 1.5 nm, for both setups.

## Data availability

The authors declare that the data supporting the findings of this study are available within the paper and its Supplementary files. The data that support the findings of this study are available from the corresponding authors upon request. Source data are provided with this paper.

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

## Acknowledgements

The authors are grateful to Dr. Sebastiano, Di Pietro, and Dr. Laura Abad Galan for the synthesis of rare earths-chiral complexes and acknowledge support from Agence Nationale de la Recherche (SMMCPL ANR-19-CE29-0012-02).

## Author contributions

S.G. and B.B. jointly conceived the idea and developed the theory. S.G., B.B., and A.B.-L. performed the experiments and data processing. L.G., F.R., and O.M. synthesized the chemical. All the authors contributed to discussions and manuscript preparation.

## Competing interests

The authors declare no competing interests.
