## [Peer Review File · Nature Communications]

Theoretical and experimental analysis of circularly polarized luminescence spectrophotometers for artifact-free measurements using a single CCD cameraREVIEWER COMMENTS

Reviewer #1 (Remarks to the Author):

The manuscript reports a CPL instrument designed on a novel concept. It uses a single CCD camera and two independent channels. The authors show experimentally and theoretically that errors and artifacts can be minimized, by different ways of separating the two polarizations. In general, this work is very well carried out, and there is a huge interest nowadays in CPL and how to measure it. I believe that all efforts leading to cheaper (with respect to commercial CPL spectrofluoropolarimeters) but reliable alternatives to measure CPL are welcome and have the potentials to expand the field even more. For these reasons, I would like to recommend the manuscript for publication after a few points are taken into account.

1) The authors are aware that a general problem of CPL, no matter how it is measured, is due to linear anisotropies of emission. Indeed, in the present set-up, such artifact comes from the coupling between S1 and QWP non-ideal retardance. The authors may elaborate on the fact that in solution, these contributions become significant where emission lifetime and molecular tumbling rate occur on a similar time scale, and therefore a depolarization due to orientational averaging can not occur completely (according to Perrin equation). When using a 90° excitation geometry, an effective way to cancel this contribution is by linearly polarizing the excitation beam on the plane of the detection. In this way only molecules that can not contribute to linear anisotropy are excited, but the cost is a relevant decrease of the signal. I suggest the authors perform the experiment by using an achiral fluorescent system giving a significant linear contribution and try to eliminate the artifacts as described above. This is important to assess, because such situation is very common when working with solid samples and rather common with organic fluorophores in solution.

2) When the polarizations are time-separated, the stability of the light source becomes critical, as the authors describe. I wonder if the instabilities observed by the authors are due to a constant drift (up or downwards) or just to random fluctuation. If the excitation lamp signal were continuously acquired as well, would it be possible to normalize constantly the signal by the excitation intensity?

3) It is not very clear to me the origin of the artifact due to wavelength mismatch and why it is expected to be proportional to the derivative of the total luminescence. In particular, in the experiments shown in Fig. 7, is this artifact expected to be symmetric with respect to the tilt angle?

4) I appreciate that with the instruments proposed by the authors g_{lum} as low as 10^{-4} can be measured. This is a lower bound, anyway, because the actual limit depends on the efficiency of the emitter, as well. Even higher g factors become difficult to measure if the output photons are simply not enough. Can the authors quantify, the limit of their instrumentation in terms of circularly polarization brightness (B_{CPL})? I think that most instruments can easily measure compounds with a B_{CPL} around $1 \text{ M}^{-1} \cdot \text{cm}^{-1}$.

5) A rough estimate of the cost of the proposed apparatus could be given to put it into context with commercial instrumentation.

6) When mixing the two enantiomers of the Eu complex to generate Fig. 8, the authors write that "Figure 8 clearly shows this gradual racemization effect". Actually, there is no racemization effect, as this would imply some chemical process, the authors are simply preparing scalemic solutions. Please rephrase this sentence.

7) Concerning NIR-CPL, the authors may be interested in two recent papers in which measurements of Er CPL around 1500 nm with simple optics are reported (10.1021/jacs.2c01134, 10.1002/anie.202208326)

8) The language of the manuscript should be revised. For example (the list is non-comprehensive): "analyzis", "cosinus", "and CPL measurements [18, 19] and ourselves [20, 21]", "by saving one spectrophotometer, as regard CPL setups based on two of theme", etc.

Reviewer #2 (Remarks to the Author):

The manuscript describes the construction of a CPL spectrometer with an unconventional design. The instrument allows rapid acquisition of CPL spectra. A paper describing an alternative design of instrument for rapid CPL detection has already been published in Nat Comm <https://www.nature.com/articles/s41467-020-15469-5>.

I'm not sure the the design outlined in the current manuscript is demonstrably superior to that reported earlier.

This is a personal opinion, but I'm not sure that instrument development manuscripts should be published in a general science journal unless the new instrument is used to provide insight into a novel scientific problem, of reasonably broad interest. I do not believe that this manuscript meets this criteria. Consequently, I would not recommend publication.

Response to Reviewers

We would like to thank the reviewers for having spent their time and efforts to provide a feedback on our manuscript entitled "Theoretical and experimental analysis of circularly polarized luminescence spectrophotometers for artifact-free measurements using a single CCD camera". We believe that the reviewers comments helped us to improve the quality of the manuscript. From the reviewers comments we noticed that some aspects of the manuscript are not well understood and we did our best to make them clearer. We have taken into account all the remarks and issues of the referees. As asked by the reviewer 1, we have added some new experimental results to estimate the contribution of the fluorescence linear anisotropy on the CPL artifacts.

Please find in the following the reviewers comments in blue and our point-by-point responses.

Reviewer 1:

The manuscript reports a CPL instrument designed on a novel concept. It uses a single CCD camera and two independent channels. The authors show experimentally and theoretically that errors and artifacts can be minimized, by different ways of separating the two polarizations. In general, this work is very well carried out, and there is a huge interest nowadays in CPL and how to measure it. I believe that all efforts leading to cheaper (with respect to commercial CPL spectrofluoropolarimeters) but reliable alternatives to measure CPL are welcome and have the potentials to expand the field even more. For these reasons, I would like to recommend the manuscript for publication after a few points are taken into account.

+ We appreciate the appreciation of Reviewer 1 about our work.

1) The authors are aware that a general problem of CPL, no matter how it is measured, is due to linear anisotropies of emission. Indeed, in the present set-up, such artifact comes from the coupling between S1 and QWP non-ideal retardance. The authors may elaborate on the fact that in solution, these contributions become significant where emission lifetime and molecular tumbling rate occur on a similar time scale, and therefore a depolarization due to orientational averaging can not occur completely (according to Perrin equation). When using a 90° excitation geometry, an effective way to cancel this contribution is by linearly polarizing the excitation beam on the plane of the detection. In this way only molecules that can not contribute to linear anisotropy are excited, but the cost is a relevant decrease of the signal. I suggest the authors perform the experiment by using an achiral fluorescent system giving a significant linear contribution and try to eliminate the artifacts as described above. This is important to assess, because such situation is very common when working with solid samples and rather common with organic fluorophores in solution.

+This comment highlight a very important point. In the first version of the article, we worked only with isotropic solutions without polarization photoselection. We agree that linear anisotropies may lead to undesired measurement artifacts when the isotropy of the medium can not be ensured. We therefore determined the possible contribution of the linear anisotropies to the CPL signal from both a theoretical and an experimental point of view:

1. **We have added an experimental study on fluorescein molecules as a function of the viscosity of the solution - Section 5.** It shows that indeed a CPL signals can be measured for highly viscous solutions even if fluorescein is achiral! This founding (anticipated by Reviewer 1) makes us go deeper into the analysis of our set-up.
2. From a theoretical point of view, **we took into account circular and $\pm 45^\circ$ dichroism in the modeling of the PBS** (lines 145–150). We have included these parameters the SI, equation 2 and table 2 which gathers all the results. Finally, these additional parameters complicate the maths but explain very well the experimental results.
3. The contributions of the LD' and the CD of the PBS to the measured CPL are now quantified (table 2). We choose not to add the consideration of the linear anisotropies for the description of the space separation set-up while in any event it is not the best to perform the CPL measurement. For the time and the time+space set-up, only the residual CD in the PBS induces a CDS_1^e signal that can not be differentiated from the S_3^e one's. It is added in Eq.4 and 6 and discuss accordingly.
4. In parallel, we have recorded the $\pm 45^\circ$ and CD of the polarizing beam splitter with our CD/LD spectrometer (SI Section 3). It reveals an unexpected few percents of signals for the circular and $\pm 45^\circ$ linear dichroism. This value of PBS circular dichroism is in very good agreement with the theory and experimental result on the fluorescein.

Thanks again to Reviewer 1, for pointing out what was missing for a complete analysis.

2) When the polarizations are time-separated, the stability of the light source becomes critical, as the authors describe. I wonder if the instabilities observed by the authors are due to a constant drift (up or downwards) or just to random fluctuation. If the excitation lamp signal were continuously acquired as well, would it be possible to normalize constantly the signal by the excitation intensity?

Within our integration time the random fluctuation is very small and the signal difference comes mainly from the drift. In Sec. 3.2 (line 303), we estimate this drift around 1%. Theoretically we agree Reviewer 1 : if we are able to continuously measure the pump power, we can normalize the CPL signal. However, from an experimental point of view, we need a measurement set-up with an accuracy $\sim \frac{1\%}{10} = 0.1\%$. Standard photodiode sensors (thorlabs- Photodiode Power Sensors) which combine enough speed and accuracy are proposed with a linearity of $\pm 0.5\%$ and an accuracy of $\pm 5\%$ which make them useless. Trying to normalize the signal with the recorded pump power will therefore report the issue from the stability of the pump to the stability of the measurement set-up and require a rather sophisticated detection apparatus with thermally controlled electronic. We like better rely on the two measurements set-up which reveals to be very efficient.

To explain this point in the paper we have added line 303: *These intensity variations come from the slowly varying drift of the pump source. To monitor this drift in order to scale the fluorescence intensity would require detection set-up with an accuracy of $\approx 0.1\%$. This can not be performed with standard photodiode power sensors which usually present a few percent accuracy.*

3) It is not very clear to me the origin of the artifact due to wavelength mismatch and why it is expected to be proportional to the derivative of the total luminescence. In particular, in the experiments shown in Fig. 7, is this artifact expected to be symmetric with respect to the tilt angle?

We now explain this in the paper line 201 after “Less intuitive, the wavelength mismatch of the two recorded spectra induces a CPL signal proportional to the luminescence derivative ($\frac{\partial S_0}{\partial \lambda}$): “ Indeed the difference between two *identical* spectra recorded with a wavelength mismatch $\Delta\lambda$ writes: $\Delta S = S(\lambda + \Delta\lambda) - S(\lambda)$. Because $\Delta\lambda \ll \lambda$, we may use the Taylor-series expansion: $S(\lambda + \Delta\lambda) \approx S(\lambda) + \Delta\lambda \frac{dS}{d\lambda}(\lambda)$. Therefore, the recorded CPL signal, being a difference between two unmatched signals, contains unwanted terms proportional to the signal derivative.”

In the experiment shown in Fig. 7, we tilted the camera in order to experimentally add a wavelength mismatch between the top and bottom spectra recorded on the camera. The goal is to qualitatively check the theory not to do quantification. Indeed, the exact wavelength mismatch is wavelength dependent and is quite complicated to model because it depends on the position of the rotation center, the exact dispersion formula and the optical aberrations at the high aperture angles.

However as Reviewer 1 mentioned, by inverting the tilt angle, the mismatch is inverted and the artifact also (false $\text{CPL} \propto \Delta\lambda \frac{\partial S_0}{\partial \lambda}$). We now describe this in the new version (line 328): “ These artifacts are especially intense in the 620 nm region where the fluorescence derivative is the highest. At a maximum camera tilt angle of -1.2° , the artifacts are nearly proportional to the derivative (black dashed curve). By inverting the angle to $+1.2^\circ$, the artifacts change in sign but not in shape as expected by the theory (Tab. 2 line 3, false $\text{CPL} \propto \Delta\lambda \frac{\partial S_0}{\partial \lambda}$).”

I appreciate that with the instruments proposed by the authors g_{lum} as low as 10^{-4} can be measured. This is a lower bound, anyway, because the actual limit depends on the efficiency of the emitter, as well. Even higher g factors become difficult to measure if the output photons are simply not enough. Can the authors quantify, the limit of their instrumentation in terms of circularly polarization brightness (B_{CPL})? I think that most instruments can easily measure compounds with a B_{CPL} around $1 \text{ M}^{-1} \cdot \text{cm}^{-1}$.

We agree with the remark of Reviewer 1, it is better to consider the B_{CPL} in order to compare the results. We discuss now of the capability of our set-up considering the B_{CPL} (emitter parameter) and the integration time required to get “good” spectra (signal to noise higher than 10). It was written in the last two paragraphs of the introduction, in the signal to noise ratio discussion (section 3.5) and in the conclusion.

5) A rough estimate of the cost of the proposed apparatus could be given to put it into context with commercial instrumentation. The cost of this kind of apparatus is about 40k€ mainly driven by the CCD camera (about 30k€). As requested, we have included this information in the conclusion.

6) When mixing the two enantiomers of the Eu complex to generate Fig. 8, the authors write that Figure 8 clearly shows this gradual racemization effect. Actually, there is no racemization effect, as this would imply some chemical process, the authors are simply preparing scalemic solutions. Please rephrase this sentence.

We, agree totally, as requested, the sentence as been rephrased.

7) Concerning NIR-CPL, the authors may be interested in two recent papers in which measurements of Er CPL around 1500 nm with simple optics are reported (10.1021/jacs.2c01134, 10.1002/anie.202208326)

We thank Reviewer 1 for the information. These references have been added in the introduction.

8) The language of the manuscript should be revised. For example (the list is non-comprehensive): analysis, cosine, and CPL measurements [18, 19] and ourselves [20, 21], by saving one spectrophotometer, as regard CPL setups based on two of theme, etc.

As requested, the text was thoroughly read and we made our best to rephrase or correct the language.

Reviewer 2

The manuscript describes the construction of a CPL spectrometer with an unconventional design. The instrument allows rapid acquisition of CPL spectra. A paper describing an alternative design of instrument for rapid CPL detection has already been published in Nat COmm <https://www.nature.com/articles/s41467-020-15469-5>.

I'm not sure the the design outlined in the current manuscript is demonstrably superior to that reported earlier.

We are aware of this paper and it is cited in the article. Only bright spectra of CPL Eu3+ complexes have been recorded using this setup. However, we are sorry to say that this paper presents fundamental misunderstandings. First, the calibration procedure is based on a division between two signals. While it sounds mathematically possible, it is very difficult to perform properly when the denominator signal tend to zero. They rightly use for these two signals the Eu3+ fluorescence to be sure that the beams follow the same paths. Therefore, the denominator of the calibration function is an Eu3+ spectra which has low and zero intensity values for wavelength outside the emission bands. It leads therefore to pure mathematical drawbacks: when the denominator goes to zero, the noise of the calibration function will dramatic enhance. Surprisingly, it is not visible in the presented spectra. In the SI, the authors explain that they use some signal treatment to calculate a common baseline. It makes sense for the non signal spectral zones but what about the bottom of the bands where the signal tends to zero?

We have finally decided to comment the limitation of this approach in the introduction of this manuscript by including the following sentence: *...This is almost impossible to achieve over the entire wavelength range and a calibration routine must be set up. A first strategy consists in measuring the calibration function between the two arms[1]. It is based on the division of one spectrum recorded on one of polarization encoding arm by the same spectrum recorded on the other polarization encoding arm. This mathematical procedure leads to a dramatic noise enhancement of the calibrating function when the intensity of the denominator is close to the background noise and requires complex baseline removal in order to avoid "division by zero" errors. Nevertheless, it allows fast CPL measurements on high $g_{lum} \geq 0.1$ molecules but no validation on low g_{lum} molecules has been described.*

We also give more details in the manuscript when speaking about the spatial separation of the polarization: *It is described by MacKenzie et al in [1] who divide the two signals coming from the two arms after inverting the role of one from left to right circular polarizer. Theoretically it eliminates the sources of artifacts proportional to ΔT . However, as it is based on the division of two signals, it induces high noise in the foot bands where the signal is close to the noise and special mathematical treatments are needed to avoid "division by zero" errors. Moreover, the derivative artifacts are enhanced as they appear both in the division and the subtraction of non perfectly λ matched spectra and can explained the remaining artifacts discussed in this work.*

Another very important error is the use of rotating polarizers in front of the spectrometers. Indeed spectrometers with their grating are highly polarization dependent devices. This has two consequences on the alignment/calibration procedure:

1. The described alignment procedure of the relative position of the polarizer and the QWP is based on the rotation of the polarizers and the comparison with a CPL spectrum (SI section 1) measured on a commercial instrument. It combines the polarizer/QWP and polarizer/grating orientations effects! Their procedure is useless and probably ends up with a compromise between right polarization orientation and arms balances. Relative orientation between QWP and polarizer can be easily done with two cross polarizers with an accuracy of about 0.05° .
2. The calibration function to balance the arms (SI Section 2) uses two orientations of one polarizer on one arm to compute the calibration factor. It is false because by changing the orientation of the polarizer the response of the arms is also modified via the monochromator response.

This last point is added at the end of the theoretical study: *A final important point is that the arm inversion must be done without changing the optical property of the arms except for the polarization encoding. It is therefore imperative not to change the polarization orientation just before the monochromator because the gratings are very polarization sensitive and with a complicated wavelength dependence. It is why we use a rotating waveplate followed by a fixed PBS: in this configuration each detector has the same response whatever the type of polarization encoding and the auto-compensation is valid. Inverted the polarizing elements (fixed QWP and rotating polarizers) results in erroneous calibration and measurement stages.*

This is a personal opinion, but I'm not sure that instrument development manuscripts should be published in a general science journal unless the new instrument is used to provide insight into a novel scientific problem, of reasonably broad interest. I do not believe that this manuscript meets this criteria. Consequently, I would not recommend publication.

We think that our set-up could allow the development of a new apparatus for measuring fast CPL measurement which is a topic in exponentially growth. All the ways to get an easier detection of CPL could have a high impact from chemical monitoring in pharamacology to in-situ biological measurements. All the “simple” CPL set-ups published earlier deal with high $B_{\text{CPL}} \geq 0.5\text{M}^{-1}\text{cm}^{-1}$. Here, we show that it can be also used for low B_{CPL} molecules as the camphorequinone providing the auto-compensation procedure. It opens the use of CPL detection to wide range of molecules. This point is added in the introduction (line 63): *All these results have been obtained with emitters having CPL brightness factors $B_{\text{CPL}} \geq 0.5\text{M}^{-1}\text{cm}^{-1}$ [2]. In this study, we show that a single scan CPL setup can be extended for the measurement of CPL spectra of chiral compounds whatever their CPL brightness.*

References

- [1] Lewis E. MacKenzie, Lars-Olof Pålsson, David Parker, Andrew Beeby, and Robert Pal. Rapid time-resolved circular polarization luminescence (cpl) emission spectroscopy. *Nature Communications*, 11(1):1676, April 2020.
- [2] Lorenzo Arrico, Lorenzo Di Bari, and Francesco Zinna. Quantifying the overall efficiency of circularly polarized emitters. *Chemistry – A European Journal*, 27(9):2920–2934, 2021.

REVIEWERS' COMMENTS

Reviewer #1 (Remarks to the Author):

I thank the authors for their careful revisions. They have thoroughly revised the manuscripts by adding new important data and addressing all my points. I think that now the work is complete and of excellent quality. I therefore recommend publication as is.